# NGS-Based Identification of Two Novel *PCDH19* Mutations in Female Patients with Early-Onset Epilepsy

**DOI:** 10.3390/ijms25115732

**Published:** 2024-05-24

**Authors:** Renata Szalai, Kinga Hadzsiev, Agnes Till, Andras Fogarasi, Timea Bodo, Gergely Buki, Zsolt Banfai, Judit Bene

**Affiliations:** 1Department of Medical Genetics, University of Pecs Medical School, 7624 Pecs, Hungary; szalai.renata@pte.hu (R.S.); hadzsiev.kinga@pte.hu (K.H.); till.agnes@pte.hu (A.T.); buki.gergely@pte.hu (G.B.); banfai.zsolt@pte.hu (Z.B.); 2Child Neurology Department, Bethesda Children’s Hospital, 1146 Budapest, Hungary; fogarasi.andras@bethesda.hu (A.F.); bodo.timea@bethesda.hu (T.B.); 3Andras Peto Faculty, Semmelweis University, 1125 Budapest, Hungary

**Keywords:** *PCDH19* mutation, epilepsy, protocadherin, NGS, WES

## Abstract

Developmental and epileptic encephalopathy-9 (DEE9) is characterized by seizure onset in infancy, mild to severe intellectual impairment, and psychiatric features and is caused by a mutation in the *PCDH19* gene on chromosome Xq22. The rare, unusual X-linked type of disorder affects heterozygous females and mosaic males; transmitting males are unaffected. In our study, 165 patients with epilepsy were tested by Next Generation Sequencing (NGS)-based panel and exome sequencing using Illumina technology. *PCDH19* screening identified three point mutations, one indel, and one 29 bp-long deletion in five unrelated female probands. Two novel mutations, c.1152_1180del (p.Gln385Serfs*6) and c.830_831delinsAA (p.Phe277*), were identified and found to be de novo pathogenic. Moreover, among the three inherited mutations, two originated from asymptomatic mothers and one from an affected father. The *PCDH19* c.1682C>T and c.1711G>T mutations were present in the DNA samples of asymptomatic mothers. After targeted parental testing, X chromosome inactivation tests and Sanger sequencing were carried out for mosaicism examination on maternal saliva samples in the two asymptomatic *PCDH19* mutation carrier subjects. Tissue mosaicism and X-inactivation tests were negative. Our results support the opportunity for reduced penetrance in DEE9 and contribute to expanding the genotype–phenotype spectrum of *PCDH19*-related epilepsy.

## 1. Introduction

*PCDH19* (Protocadherin 19)-related childhood-onset epilepsy syndrome, recently named epileptic encephalopathy, early infantile, type 9, developmental and epileptic encephalopathy-9 (DEE9) or Juberg-Hellmann syndrome (OMIM#300088), is a female-restricted, X-linked disorder with an extraordinary male-sparing inheritance pattern (affects heterozygous females and mosaic males only; hemizygous males are unaffected) and an estimated 4.85 per 100,000 live born females [1,2].

The phenotype of patients with a pathogenic mutation in the *PCDH19* gene is variable. In the literature, there is no definitely determined genotype–phenotype correlation between the location or type of variant and the phenotypic features, like seizure onset [3,4]. Previous results suggest that missense variants may be more likely to be associated with a less severe phenotype compared to loss-of-function variants and whole-gene deletions [5]. DEE9 is characterized as an early-onset seizure disorder triggered by fever, resembling Dravet syndrome (DS). Although the clinical spectrum of *PCDH19* mutation-positive patients can show an overlap with DS patients, the age at onset, the occurrence of status epilepticus, the absence of seizures, myoclonic seizures, and the long-term outcome are different. The most frequent clinical features in *PCDH19*-associated epilepsy are seizures (with various types and times of onset), often associated with fever, mild to profound intellectual disability (ID), and psychiatric symptoms, including autistic features, attention deficit hyperactivity disorder (ADHD), anxiety, obsessive-compulsive disorder, and schizophrenia [3]. Although the majority of cases are reported with normal brain magnetic resonance imaging results, focal cortical dysplasia, hippocampal sclerosis, arachnoid cysts, and subependymal periventricular nodular heterotopia may rarely occur [6,7].

Among patients with *PCDH19*-related epilepsy, heterozygous females and a few mosaic males are affected predominantly; nonmosaic hemizygous males are asymptomatic unlike in a typical X-linked condition [8]. This unusual male-sparing inheritance pattern was first described in 1971 as a cause of epilepsy and intellectual disability limited to females without an explained molecular etiology [2]. Based on the hypothesis of Dibbens et al., since the *PCDH19* gene is subject to X-inactivation, hemizygous transmitting males are likely to have a homogeneous population of *PCDH19*-mutant cells, whereas affected females are probably mosaic, comprising two cell populations (*PCDH19*-mutant and *PCDH19*-wildtype cells) [1].

*PCDH19* (NM_001184880.2) is one of the most clinically relevant genes in epilepsy. The *PCDH19* gene is located at Xq22.1 and encodes a protocadherin 19 (*PCDH19*) transmembrane protein. Protocadherins, as a subgroup of the cadherin superfamily, play an important role in calcium-dependent cell–cell interaction and adhesion [9]. The protocadherin 19 membrane protein comprises six extracellular cadherin repeats, one transmembrane domain, and one intracellular tail [10]. The above-mentioned tissue mosaicism in females may distract cell–cell interaction, resulting in early infantile epileptic encephalopathy clinically (Figure 1). Somatic mosaicisms have been reported in unaffected or mildly affected parents, proposing the opportunity of reduced penetrance. Parental mosaic mutations of *PCDH19* are presumably underestimated [11,12,13].

Since *PCDH19* is predominantly expressed in developing and adult human nervous systems, including the hippocampus, cortex, limbic region, and endothelial cells mainly in the central nervous system, it is suggested to play a role in cognitive function [14,15]. Expression of the protein was not detected in white matter tracts. In affected patients, the impairment of cognitive function may be explained by the dysfunction of synaptic neuronal connections, signal transduction, and axon outgrowth [3].

To date, 272 *PCDH19* variants have been reported in The Human Gene Mutation Database (https://www.hgmd.cf.ac.uk/ac/gene.php?gene=PCDH19, accessed on 20 January 2024). All types of DNA variants were identified, including nonsense, missense, splice site and intergenic mutations, small deletions, insertions, and gross rearrangements [4,8]. About half of the reported mutations are de novo and have been identified mostly in females (90%). Previous studies reported that missense mutations represent about 46–58% of variants, followed by nonsense mutations and small indels [4,8]. Whole-gene deletions are rarely reported, but the frequency of this kind of alteration is probably underestimated due to the applied direct sequencing technique, which is unable to detect larger rearrangements [16,17].

The *PCDH19* gene consists of six exons. Exon 1 is the largest exon, which encodes the extracellular domain responsible for cell–cell interactions. Pathogenic missense and truncating mutations are situated in this hot spot region and modify the calcium-binding properties of the protein or result in a loss of the calcium-mediated cell–cell adhesion.

In this study, we screened 165 patients with epilepsy and found *PCDH19* mutation in 5 female patients with idiopathic Dravet-like epileptic encephalopathy and psychiatric abnormalities. The aim of our study was to characterize the detected two novel and three reported mutations in our patient cohort in order to better understand the clinical features and the mutational spectrum of *PCDH19*-related epilepsy.

## 2. Results

The detected mutations in *PCDH19* (NM_001184880.2) are summarized in Table 1. Mutation screening identified three missense point mutations, one indel resulting in a premature termination codon, and one 29 bp-long deletion. All the detected mutations are located in exon 1, encoding the extracellular domain.

In Patients 2 and 4, two pathogenic mutations were found to be de novo and have not been reported in the literature previously. Two of the three known previously described mutations were pathogenic; the third was a variant of uncertain significance (VUS).

Targeted DNA custom epilepsy Next Generation Sequencing (NGS) panel sequencing identified a previously described pathogenic *PCDH19* c.1682C>T (p.Pro561Leu) variant in Patient 1, a pathogenic c.1152_1180del (p.Gln385Serfs*6) deletion in Patient 2, which has not been reported, and a known c.1711G>T (p.Gly571Cys) VUS in Patient 3.

Regarding Patient 4, exome sequencing analysis identified a c.830_831delTCinsAA deletion/insertion resulting in a novel pathogenic p.Phe277* mutation in the *PCDH19* gene. The preliminary results of exome and Sanger sequencing showed T>A and C>A changes affecting two adjacent nucleotides at coding positions 830 and 831. However, we analyzed the reads derived from the WES using the Integrative Genomics Viewer (IGV 2.12.0) and found that the two affected bases are located on the same allele of the *PCDH19* gene (Figure 2), so it is a complex indel (delTCinsAA) mutation.

Sequence analysis of the comprehensive epilepsy panel detected a known pathogenic missense *PCDH19* c.1031C>T (p.Pro344Leu) mutation in Patient 5.

The ACMG classification of the novel *PCDH19* c.1152_1180del and c.830_831delTCinsAA mutations are pathogenic because these mutations are de novo, predicted to cause disease, and these identified variants are not found either in gnomAD genomes or in gnomAD exomes (PVS1, PM2, PS2). (https://varsome.com/, https://franklin.genoox.com/clinical-db/home, accessed on 2 December 2022).

The *PCDH19* c.1682C>T (p.Pro561Leu) and c.1031C>T (p.Pro344Leu) variants are classified as pathogenic mutations (based on multiple consistent submissions and citing articles) and are associated with Developmental and Epileptic Encephalopathy 9 (OMIM#300088) through the ClinVar database. Moreover, these variants are not found either in gnomAD genomes or in gnomAD exomes and are located in a hot spot region. Previously, pathogenic/likely pathogenic alternative variants were identified at this position and the pathogenicity prediction model MetaRNN also suggests the pathogenic classification. The classification of the detected *PCDH19* c.1711G>T (p.Gly571Cys) single nucleotide variant is VUS in ClinVar without any citation and functional evidence (https://www.ncbi.nlm.nih.gov/clinvar/, accessed on 2 December 2022).

After parental targeted tests, only one case (Patient 5) showed the classic paternal transmission of the mutation from the asymptomatic father (Figure 3). Novel mutations were found to be de novo. Interestingly, two identified mutations (*PCDH19* c.1682C>T and c.1711G>T) were also observed in asymptomatic mothers (of Patients 1 and 3). Thus, to clarify the pathogenic role of these variants, mosaicism analysis, X chromosome inactivation tests, and exome sequencing were performed.

The circles represent females and the squares represent males. The filled symbols indicate female family members with epilepsy. A dot denotes asymptomatic *PCDH19* hemizygous male family members. The proband (Patient 5) is marked with an arrow (III:2).

Sanger sequencing on maternal saliva samples did not detect tissue mosaicism (Figure 4).

X-inactivation tests resulted in a slightly skewed ratio from the random (50:50) inactivation in Patient 1 and her mother. Moreover, although the X chromosome inactivation was slightly skewed to the mutant allele in the sample of Patient 1’s mother (60:40), it was slightly skewed to the normal allele in the sample of Patient 1 (33:67) (Figure 5).

Since the obtained results did not strengthen our hypothesis in Patients 1 and 3, whole-exome sequencing was carried out in order to elucidate the pathological role of the detected mutations. In Patient 1, exome sequencing analysis did not identify any additional variant that may be responsible for the phenotype. In Patient 3, WES identified a further heterozygous missense VUS in the *KCNQ5* (NM_001160133.1) gene. The *KCNQ5* c.2291C>A (p.Pro764Gln) variant would match the phenotype, but it was also detected in the maternal DNA sample. No other variants were identified that are compatible with the phenotype of Patients 1 and 3.

Subsequent seizure types were observed: epileptic spasms, generalized tonic–clonic seizure (GTCS), complex partial, myoclonic, atypical absence, status epilepticus, and clonic seizures. The age of seizure onset was from 11 months to 5 years. We found mild intellectual deficits and developmental delays in our patients. All the patients were characterized by different psychiatric abnormalities, such as autistic features, impulsive behavioral problems, and ADHD. Associated brain abnormalities, including enlarged ventricles, corpus callosum hypoplasia, partial hippocampal sclerosis, and caudate nucleus lesions were described in our patients. In Patients 1, 4, and 5, additional minor, non-characteristic phenotypic anomalies were observed (Table 2).

With respect to Patient 1, it is surprising that the asymptomatic mother is a heterozygous *PCDH19* mutation carrier since the mutation identified and passed on to her epileptic daughter is a known variant with pathogenic clinical significance.

Patient 3 proved to be the most interesting case, with an asymptomatic carrier mother and unusual traits in *PCDH19*-related epilepsy disorder, including the particular age at disease onset (5 years) and the absence of fever as a provoking factor and in whom we detected an additional VUS in the *KCNQ5* gene.

Only one case, Patient 5, initially clinically thought to have Dravet syndrome (she presented fever-triggered seizures in the first year of life, status epilepticus, behavioral problems, and developmental delay after normal infant development), was found to have *PCDH19*-related familial inheritance and expected phenotypic manifestation based on the pedigree (Figure 3) [16].

## 3. Discussion

*PCDH19*-related epileptic encephalopathy is an X-linked disorder with an unusual inheritance [4]. Associated phenotypes are characterized by seizures, intellectual disability, and behavioral problems. The phenotypic features have a spectrum, ranging from well-controlled epilepsy with normal cognitive development to intractable epilepsy with severe intellectual impairment [1,16,18,19,20,21,22]. Clinical features associated with *PCDH19* mutations may overlap with Dravet syndrome [8,16,21]. When female patients phenotypically resemble DS but *SCN1A* genotyping tests prove to be negative, *PCDH19* mutation screening should be under consideration [23]. Previous studies have demonstrated that 16% of DS patients carry deleterious variants in the *PCDH19* gene [24,25]. Therefore, for the DS-like phenotype, the genetic testing of the *PCDH19* gene would be important after a negative *SCN1A* result.

The prevalence of developmental and epileptic encephalopathy type 9 is presumably underestimated. Firstly, clinical recognition can be difficult because of the unique inheritance pattern, the lack of a family history of seizures, or the overlapping phenotype. Additionally, epilepsy disorder is often underdiagnosed due to the method used or the influence of other factors. The majority of studies carried out solely used Sanger sequencing to screen *PCDH19*, which is unable to identify gross heterozygous deletions.

In this study, we report five female patients with Dravet-like epileptic encephalopathy, cognitive impairment, and psychiatric features. Our results are partly consistent with previous findings from other researchers. A wide range of epileptic seizure types from absence to status epilepticus with different EEG abnormalities was found. All the patients were characterized with cognitive impairment. Each patient demonstrated various psychiatric abnormalities, as commonly described in *PCDH19*-related epilepsy disorder.

Regarding inheritance, in our patient cohort, the transmission of the mutations occurred via maternal transmission in the case of Patients 1 and 3 and through paternal transmission in Patient 5. Novel mutations were found to be de novo in Patients 2 and 4 (Table 1). Interestingly, we found in one case (in the family of Patient 5) the characteristic *PCDH19*-related familiar inheritance pattern with paternal transmission. Typically, it was found in the studied family that there was an asymptomatic hemizygous father and a symptomatic heterozygous paternal female cousin beside an affected pathogenic *PCDH19* mutation carrier female proband. As a result of a targeted parental carrier examination, we found that the carriers of the *PCDH19* mutation were the asymptomatic mothers of Patients 1 and 3. This is even more surprising as the clinical significance of the identified variant in a DNA sample of Patient 1 is pathogenic according to the ClinVar database. In female patients, somatic mosaicism or skewed X-inactivation may explain the unusual carrier test results. Since the *PCDH19* mutation carrier mothers of Patients 1 and 3 were asymptomatic, in view of the discrepant findings, we carried out methylation profile-based X-inactivation tests for the examination of presumed non-random X-inactivation, and Sanger sequencing was performed on maternal saliva samples to examine potential tissue mosaicism. We could not detect either significant skewing in X-inactivation or tissue mosaicism. One possible explanation is that the X chromosome inactivation ratio (the extent of skewing) may differ in distinct tissues (peripheral blood vs. neuronal tissue). A further explanation may be that the age at X chromosome inactivation testing varied in our study, which may also affect the comparability between the probands and their mothers [26].

Finally, due to the negative X chromosome inactivation and tissue mosaicism results, we carried out WES to find further causative variants in Patients 1 and 3. In the sequencing data from the DNA of Patient 1, there was no other causative variant that would have been compatible with the phenotype. In contrast, for Patient 3, from exome sequencing data, an additional VUS (c.2291C>A; p.Pro764Gln) was identified in *KCNQ5* (NM_001160133.1). The *KCNQ5* gene is associated with autosomal dominant intellectual developmental disorder-46 (OMIM#617601). Based on clinical information, the detected variant may be considered to explain or contribute to the phenotype, in particular, the ID and seizures. This variant was also detectable from the DNA sample of the mother of Patient 3. After the obtained results of WES, the detected *KCNQ5* c.2291C>A (p.Pro764Gln) and *PCDH19* c.1711G>T (p.Gly571Cys) variants of uncertain significance are likely to play a role together in the development of the phenotype and are responsible for the symptoms of Patient 3. Although Patient 3 is considered atypical among patients with *PCDH19*-associated epilepsy, the detected *PCDH19* mutation may explain some of her symptoms, such as cognitive regression, as a consequence of epileptic encephalopathy.

There are several possible reasons that could explain the carrying of a pathogenic *PCDH19* mutation by an asymptomatic mother. Previously, in the literature, Depienne et al. reported that 7% (3/44) of *PCDH19* mutations are inherited from asymptomatic mothers [8]. The fact that an asymptomatic mother is carrying a pathogenic *PCDH19* mutation can also be explained by the fact that among females with a heterozygous *PCDH19* mutation, penetrance is incomplete (approx. 90%) [11]. Although a thorough clinical examination of the patients was performed prior to genetic testing, in the asymptomatic mothers carrying the pathogenic *PCDH19* mutation, the possibility of an unrecognized and, thus, undocumented mild epilepsy in the mother’s childhood cannot be excluded.

The novelty of our findings is the two newly identified mutations in the *PCDH19* gene. In Patient 2, we detected a novel 29 bp deletion (c.1152_1180del; p.Gln385Serfs*6) leading to a frameshift and, finally, to a truncated protein (Figure 6). Additionally, the c.830_831delinsAA results in a premature stop codon (p.Phe277*), thus forming a truncated protein, which has also not been reported in the literature. The identification of c.830_831delinsAA was carried out through WES in Patient 4. The NGS-based sequencing technique, like WES, allows the identification of this kind of rare complex sequence alteration and the determination of the allelic origin of the affected adjacent nucleotides.

## 4. Materials and Methods

### 4.1. Patient and Sample Recruitment

Between 2020 and 2022, 165 patients with epilepsy were tested using NGS-based techniques (epilepsy gene panel (*SCN1A*, *SCN8A*, *ARX*, *CHD2*, *PCDH19*, *SLC2A1*, *STXBP1)*, a comprehensive epilepsy gene panel containing 379 nuclear-encoded genes, and exome sequencing) in our laboratory. Of these, five female patients with a spectrum of epilepsy phenotype and *PCDH19* mutation were enrolled in this study. Table 2 summarizes the clinical characteristics of these five patients.

The study was approved by the Ethics Committee of the University of Pécs (Protocol 8770-PTE 2021.) After genetic counselling, written informed consent was obtained from the participating members prior to their inclusion in the study. During the collection and analysis of the DNA samples and processing of the accompanying clinical and personal data, the guidelines and regulations of the Helsinki Declaration of 1975 and the currently operative national regulations (Hungarian law; XXI/2008) were followed. Genomic DNA was extracted from peripheral blood using an E.Z.N.A. Blood DNA Maxi extraction kit (OMEGA^®^, Bio-tek, Inc., Norcross, GA, USA) and from saliva using Norgen’s Saliva DNA Isolation Reagent Kit (Norgen Biotek Corp., Thorold, ON, Canada). The concentration and purity of the extracted DNA were measured using the NanoDrop 2000 spectrophotometer (Thermo Fisher Scientific, Waltham, MA, USA).

### 4.2. Generation and Analysis of Sequence Data

For three patients (Patients 1–3), *NGS*-based targeted custom epilepsy gene panel (SCN1A, SCN8A, ARX, CHD2, PCDH19, SLC2A1, STXBP1) analysis was performed using a QIAGEN QIAseq library kit (Qiagen, Hilden, Germany) according to the manufacturer’s instructions and Illumina MiSeq sequencing technology (Illumina, San Diego, CA, USA) in paired-end 150 bp reads mode.

For identification of the pathogenic variant in Patient 5, a comprehensive epilepsy panel (containing 379 nuclear-encoded genes) was used.

Whole-exome sequencing was performed using DNA samples obtained from Patient 4 [27,28]. Exomic libraries were prepared using the Illumina DNA Prep with Exome 2.0 Plus Enrichment Kit (Illumina, San Diego, CA, USA), and sequencing was performed on an Illumina NovaSeq 6000 instrument according to the manufacturer’s protocol using paired-end 150 bp reads. The reads were aligned to the human reference genome (GRCh37:hg19) using the Burrows–Wheeler Aligner [29].

For the classification and interpretation of the genomic data, the guidelines of the American College of Medical Genetics and Genomics (ACMG) were used [30]. Moreover, databases, such as ClinVar (https://www.ncbi.nlm.nih.gov/clinvar, accessed on 15 November 2022), The Genome Aggregation Database (gnomAD) (https://gnomad.broadinstitute.org, accessed on 15 November 2022), Genomes/Exomes coverage, and in silico prediction tools, such as Mutation taster (https://www.mutationtaster.org, accessed on 15 November 2022), PhastCons and PhyloP, were used (http://compgen.cshl.edu/phastweb, accessed on 15 November 2022).

Direct sequencing was performed to detect the presence of variants identified by the NGS panel and exome sequencing. Moreover, targeted parental Sanger sequencing was carried out using an ABI 3500 Genetic Analyzer (Applied Biosystems, Foster City, CA, USA) [31].

### 4.3. X-Inactivation Studies

X-inactivation study and an estimation of the degree of skewing were performed as previously reported by Lau et al. [32]. Genomic DNA from blood was digested overnight at 37 °C with the HpaII enzyme (Roche Diagnostics, Mannheim, Germany) [33,34]. After digestion and enzyme inactivation, a PCR reaction was performed on both the digested and undigested samples with fluorescent forward (5′-FAM-TCCAGAATCTGTTCCAGAGCGTGC-3′) and reverse (5′-GCTGTGAAGGTTGCTGTTCCTCAT-3′) primers specific for the androgen receptor gene. The PCR was analyzed on the ABI 3130 DNA sequencer (Applied Biosystems, Foster City, CA, USA). X-inactivation was classified as random (ratio 50:50 to 80:20) or skewed (ratio > 80:20) [20,32].

### 4.4. Mosaicism Determination

Sanger sequencing was performed on maternal DNA samples isolated from saliva to examine possible tissue mosaicism using an ABI 3500 Genetic Analyzer (Applied Biosystems, Foster City, CA, USA).

## 5. Conclusions

In conclusion, genetic testing of *PCDH19* as the second most important gene in patients with fever-provoked, intractable, infantile epileptic encephalopathy could play a crucial role in the therapy and prognosis of epilepsy. In *PCDH19*-related epilepsy, genetic counselling and clinical geneticists play an essential role in the recognition of the disease, the selection of the appropriate diagnostic test, and the identification of affected family members. It can be a diagnostic challenge for the clinician since developmental and epileptic encephalopathy-9 is a disease characterized by an unusual inheritance pattern, variable manifestations, and severity that is additionally influenced by various factors including epigenetic mechanisms, incomplete penetrance, and mosaicism, which can further complicate the clinical picture.

Genetic testing of *PCDH19* may be useful for all patients with Dravet-like epileptic encephalopathy who have a negative test for *SCN1A*. Accurate knowledge of the responsible gene and genetic variant can significantly influence the choice of therapy. Female patients should also be tested in the presence of the following: early-onset, idiopathic, generalized epilepsy, focal onset seizure, mild epilepsy in remission, irrespective of MRI findings, fever, cognitive impairment, and positive family history. Further genotype–phenotype examinations and functional studies of the nervous system are needed to better understand the etiology of the disease and clarify unacknowledged questions.

## Figures and Tables

**Figure 1 ijms-25-05732-f001:**
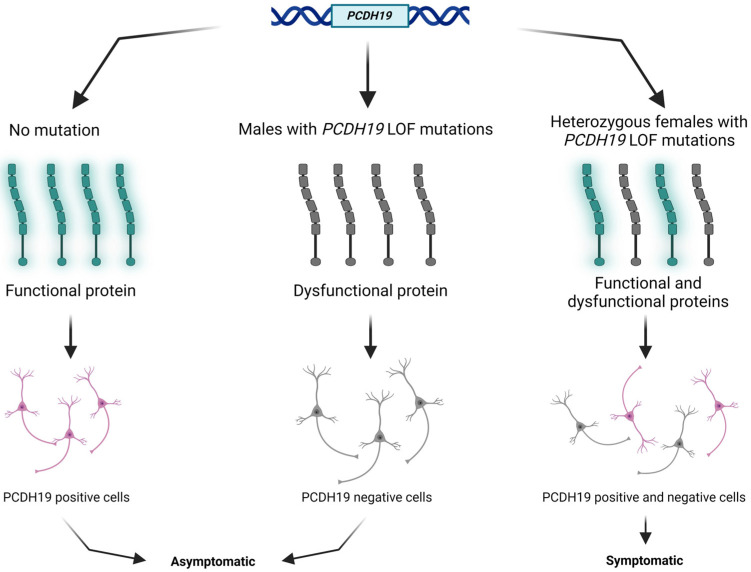
Schematic presentation of the cellular interference mechanism with *PCDH19* mutation and in normal individuals. The figure was created using BioRender.com (accessed on 16 April 2024).

**Figure 2 ijms-25-05732-f002:**
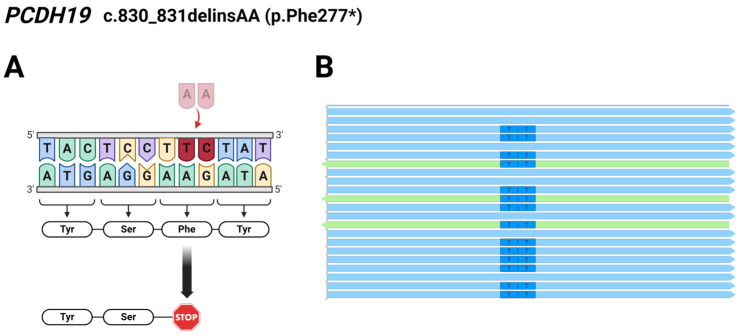
(**A**) Schematic illustration of the identified pathogenic *PCDH19* c.830_831delTCinsAA (p.Phe277*) mutation in Patient 4; The burgundy coloured TC nucleotides were replaced by a pale pink coloured AA nucleotides. (**B**) Reads obtained from the exome sequencing analyzed using the Integrative Genomics Viewer (2.12.0) show the two affected bases located on the same allele of *PCDH19*. Light blue arrows represent forward reads, green arrows show reverse reads. The figure was created using BioRender.com (accessed on 16 April 2024).

**Figure 3 ijms-25-05732-f003:**
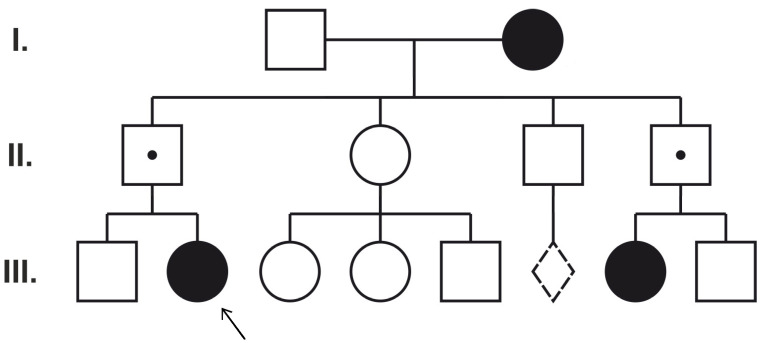
Pedigree of Patient 5. Dotted line diamond represents miscarriage. I, II, III numbers show different generations.

**Figure 4 ijms-25-05732-f004:**
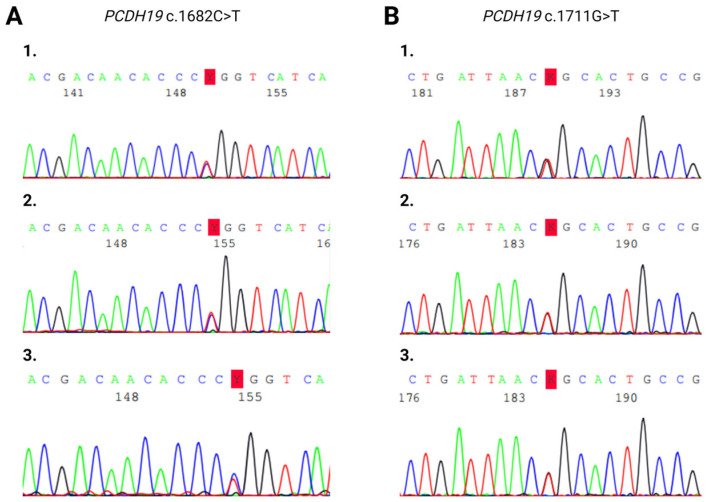
Sanger sequencing electropherogram of heterozygous *PCDH19* (**A**) c.1682C>T and (**B**) c.1711G>T mutations. (**A/1**) Blood sample of Patient 1. (**A/2**) Blood sample of Patient 1’s mother. (**A/3**) Saliva sample of Patient 1’s mother. (**B/1**) Blood sample of Patient 3. (**B/2**) Blood sample of Patient 3’s mother. (**B/3**) Saliva sample of Patient 3’s mother.

**Figure 5 ijms-25-05732-f005:**
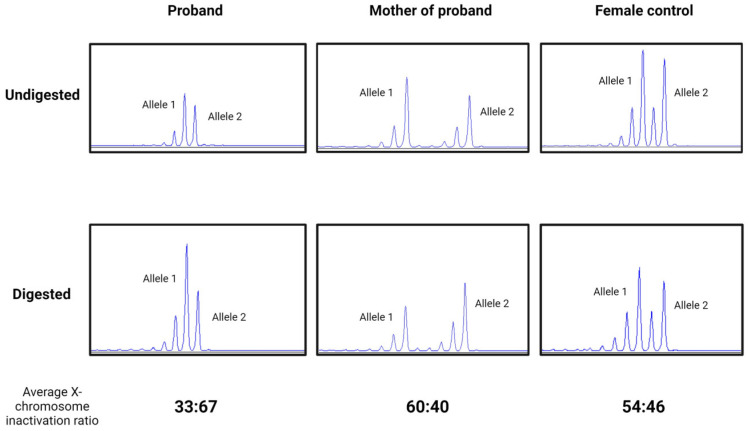
Electropherogram of X-inactivation tests in Patient 1, her mother, and a female control in undigested and predigested samples using methylation-specific restriction endonuclease *HpaII*. A slightly skewed ratio in Patient 1 (proband) (33:67), in her mother (60:40), and a random (54:46) X-inactivation pattern in a healthy female control.

**Figure 6 ijms-25-05732-f006:**
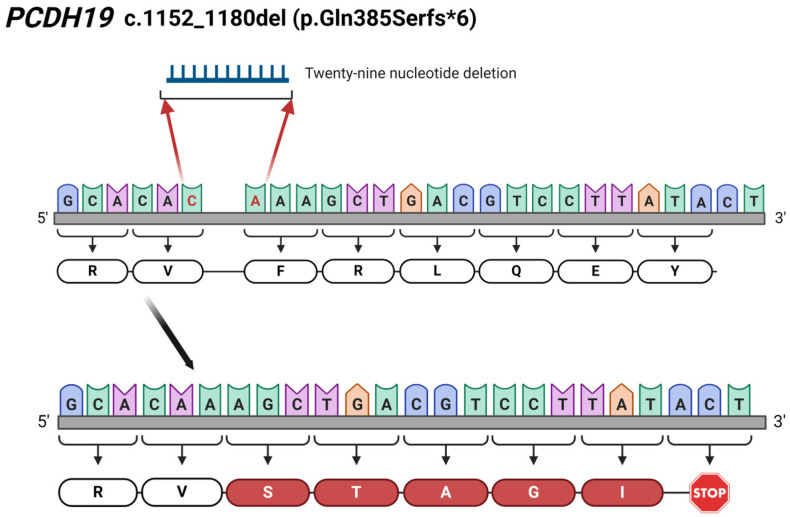
Illustration of the identified 29 bp-long c.1152_1180del (p.Gln385Serfs*6) deletion in *PCDH19* gene in Patient 2. The red coloured C and A nucleotides are the first and last nucleotide involved in the 29 bp-long deletion. The figure was created using BioRender.com (accessed on 16 April 2024).

**Table 1 ijms-25-05732-t001:** Summary of the detected *PCDH19* mutations in five patients with epilepsy.

*PCHD19* Mutation	Patients				
	P1	P2	P3	P4	P5
DNA variant	c.1682C>T	c.1152_1180del ^#^	c.1711G>T	c.830_831delinsAA ^#^	c.1031C>T
Protein	p.Pro561Leu	p.Gln385Serfs*6	p.Gly571Cys	p.Phe277*	p.Pro344Leu
Location	Exon 1	Exon 1	Exon 1	Exon 1	Exon 1
Genotype	heterozygous	heterozygous	heterozygous	heterozygous	heterozygous
Type	missense	deletion (29bp)	missense	nonsense	missense
ACMG classification	Pathogenic (PM1, PM2, PM5, PP3, PP5)	Pathogenic (PVS1, PM2, PS2)	VUS (PM1, PM2, PP3)	Pathogenic (PVS1, PM2, PS2)	Pathogenic (PM1, PM2, PM5, PP3, PP5)
Examined family member (genotype)	mother (heterozygous) father (normal)	mother (normal) father (normal)	mother (heterozygous) father (normal)	mother (normal) father (normal) sister (normal)	paternal female cousin (heterozygous), father (hemizygous)
Transmission	maternal	de novo	maternal	de novo	paternal
Method	Epilepsy NGS panel ^¥^	Epilepsy NGS panel	Epilepsy NGS panel	WES	Comprehensive epilepsy NGS panel ^£^

VUS: Variant of uncertain significance; NGS: Next Generation Sequencing; WES: Whole Exome Sequencing; ACMG: American College of Medical Genetics and Genomics; PM1: Located in a mutational hot spot and/or critical and well-established functional domain without benign variation; PM2: Absent from controls in Exome Sequencing Project, 1000 Genomes or ExAC; PM5: Novel missense change at an amino acid residue where a different missense change determined to be pathogenic has been seen before; PS2: De novo in a patient with the disease; PP3: Multiple lines of computational evidence support a deleterious effect on the gene or gene product (conservation, evolutionary, splicing impact, etc.); PP5: Reputable source recently reports variant as pathogenic but the evidence is not available to the laboratory to perform an independent evaluation; #: not reported; ¥: (*SCN1A*, *SCN8A*, *ARX*, *CHD2*, *PCDH19*, *SLC2A1*, *STXBP1*); £: 379 nuclear-encoded genes.

**Table 2 ijms-25-05732-t002:** Clinical features of epilepsy probands with PCDH19 mutations.

Clinical Features	Patients				
	P1	P2	P3	P4	P5
Sex	F	F	F	F	F
Age at time of study	12 years	13 years	16 years	13 years	6 years
Age at onset	18 months	22 months	5 years	11 months	11 months
Family history	-	+	+	-	+
Seizure type	tonic, epileptic spasms	tonic, GTCS	GTCS, tonic, complex focal, myoclonic, atypical absence	tonic, absence, GTCS	status epilepticus, focal tonic, clonic
Provoking factors	fever, sleeping	fever, sleeping	sleeping, nutrition components, starvation, repletion, tiredness	fever, infection	fever, infection
EEG	Bilateral synchronous and asynchronous occipital IEDs	Interictal and ictal frontopolar epileptiform discharges	ESES, slow background activity, multifocal and generalized IEDs, seizures from the centro-temporo-parietal region	Postictal slow background activity	Postictal slow background activity
Cognitive impairment	mild ID, speech abnormality	mild ID	mild ID, learning difficulty	mild ID, speech delay	developmental delay
Psychiatric abnormality	behavioral problems	ADHD	cognitive regression	autism spectrum disorder	impulsive behavior
MRI findings	enlarged ventricles, corpus callosum hypoplasia	partial hippocampal sclerosis	caudate nuclei lesion	negative	negative
Others	synophrys, arched eyebrows, sacral hypopigmentation, crowded teeth	-	-	high-arched palate, down slanting palpebral fissures, synophrys, hypertrichosis	generalized hypotonia

P: Patient; F: Female; ESES: Electrical status epilepticus in sleep; GTCS: Generalized tonic–clonic seizure; ID: Intellectual disability; ADHD: Attention deficit hyperactivity disorder; EEG: Electroencephalogram; MRI: Magnetic resonance imaging; IED: Interictal epileptiform discharges.

## Data Availability

The data presented in this study are available upon request from the corresponding author.

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
