# Peer review of "NGS-Based Identification of Two Novel PCDH19 Mutations in Female Patients with Early-Onset Epilepsy"

_ijms, 2024, doi:10.3390/ijms25115732_

Round 1

Reviewer 1 Report

Comments and Suggestions for Authors

I have read the manuscript entitled 'NGS-based identification of two novel PCDH19 mutations in female patients with early-onset epilepsy” by Szalai et al. with great interest. It is a well-written and important study that describes more PCDH19 variants of a relatively small group of mutations. The methods are sound and the results are straightforward. The Tables and Figures are informative and well designed. I suggest only a few minor additions to the revised manuscript. Some methodological details and a few references should be added to complete this otherwise excellent manuscript.

Minor problems:

1) Line 2, and more: NGS; abbreviations should be spelled out when first used.

2) Line 33: Perhaps the abbreviations should be spelled out here.

3) Line 111: In Table 1, some further abbreviations should be explained (ACGM, PM1-PM5).

4) Line 219: For Dravet syndrome, please give a background and cite a reference such as Ref. 15.

5) Lines 309 and 333: The epilepsy panel, the selection of genes, etc. should be described in detail and referenced.

6) Line 321: The city and the country should be added.

7) Lines 329-331: I believe that the procedure needs further details and references should be given.

8) Lines 305 and 327: These important subsections have no references at all; please add at least a few relevant ones.

9) Lines 335-336: “Twist Human Core Exome Kit Library Prep Kit” should be referenced and the manufacturer (source) added.

10) Line 338: The human reference genome and the Burrows-Wheeler Aligner should be referenced.

11) Lines 339-340: The guideline should be linked or referenced.

12) Lines 340-342: Databases and tools should be linked and referenced (not everyone is an expert…).

13) Lines 344-345 and 359: The method and instrument used should be referenced and the maker of the analyzer should be added.

14) Line 350: The rationale using the HpaII enzyme (its activity-dependence on unmethylated sites) should be explained or referenced.

15) Line 352: Primer sequences and references should be added.

Author Response

Dear Reviewer,

We appreciate the constructive and valuable comments. We also thank you for taking the time to review our manuscript. In the revised version of the manuscript, changes are highlighted in red text. Attached please find our responses for all your comments.

Reviewer 2 Report

Comments and Suggestions for Authors

The current manuscript uses NGS to identify two novel mutations in female patients with developmental and epileptic encephalopathy-9 (DEE9), which is characterised with early onset epilepsy.

I have some comments that would strengthen the article:

General comment:

The abstract mentions developmental and epileptic encephalopathy-9 (DEE9), however, there is no other mention of DEE9 in the rest of the article. Could the authors expand upon DEE9 in the beginning of the introduction and place it in the context of the study?

Introduction

Although PCDH19 is described in (line 60 to 63) it would be better to describe this when PCDH19 Is first mentioned in line 36.

Can the authors clarify what is meant by “an extraordinary inheritance pattern” on line 38?

Line 40: are there any genotype-phenotype correlations with the PCDH19 gene i.e. are some pathogenic mutations expected to be more clinically severe? Also, could the authors clarify whether DEE9 is genetic diagnosis or whether the diagnosis can be clinical i.e., based on diagnostic criteria rather than a molecular one. I understand that some information on genotype-phenotype correlations is alluded in line 95 - 96, but I think that this information might be better placed in line 40.

It is interesting to note that most of the reported pathogenic mutations are considered as de novo (line 86). However, out of the 5 patients assessed, only two (P2 and P4) were de novo. I would have expected more given the predicted frequency of the reported pathogenic mutations. Can the authors comment on this?

Tables should be able to be interpreted by themselves. As it stands some parts of Table 1 would benefit from some further information that would help in its understanding. More information on the method (last row of Table 1) would be useful i.e., (I) epilepsy NGS panel vs. comprehensive epilepsy NGS panel and (II) the utility of using both ClinVar and ACMG classification. Would it not have been better to use ACMG classification to begin with given the clinical significance in ClinVar in two patients (P2 and P4) was not reported? Please also abbreviate ACMG in the footer of the table.

Why was a different method (WES) used for patient 4 in comparison to the other patients? Was WES a better method than NGS epilepsy/comprehensive epilepsy panel for determining the pathology of the detected mutations in the presence negative results (X chromosome inactivation and tissue mosaicism)? If so, this or similar wording could be mentioned as a note in the footer of Table 1.

In Table 2, please say what – denotes. For example, for MRI findings does the – indicate that there were no clinically significant MRI findings for P4 and P5?

Patient 5 had cognitive regression and interestingly this was the only patient with later onset (5 years). Is there any significance to this? Could this regression also be due to epileptic encephalopathy?

Discussion: If most of the reported pathogenic mutations are considered as de novo (line 86) it seems contrary to the sentence in line 224 “PCDH19-related epileptic encephalopathy is an X-linked disorder with an unusual inheritance”. Please consider amending or providing further clarification.

Conclusion: Line 361, please can the authors better clarify what is meant by “genetic testing of PCDH19 as the second most important gene in patients with epilepsy.” There are several important genes involved the pathogenesis of epilepsy in different disorders and I feel this statement is rather subjective with supporting evidence from the literature.

Author Response

(The authors gave the same response as above.)

Round 2

Reviewer 2 Report

Comments and Suggestions for Authors

Thank you for improving the manuscript.